# Metal-free photoanodes for C−H functionalization

Junfang Zhang[1,2,3], Yuntao Zhu[1], Christian Njel[4], Yuxin Liu[1,2],
Pietro Dallabernardina[1], Molly M. Stevens [3], Peter H. Seeberger[1,2],
Oleksandr Savateev [1,5] ✉ & Felix F. Loeffler [1] ✉

Organic semiconductors, such as carbon nitride, when employed as powders, show attractive photocatalytic properties, but their photoelectrochemical performance suffers from low charge transport capability, charge carrier recombination, and self-oxidation. High film-substrate affinity and well-designed heterojunction structures may address these issues, achieved through advanced film generation techniques. Here, we introduce a spin coating pretreatment of a conductive substrate with a multipurpose polymer and a supramolecular precursor, followed by chemical vapor deposition for the synthesis of dual-layer carbon nitride photoelectrodes. These photoelectrodes are composed of a porous microtubular top layer and an interlayer between the porous film and the conductive substrate. The polymer improves the polymerization degree of carbon nitride and introduces C-C bonds to increase its electrical conductivity. These carbon nitride photoelectrodes exhibit state-of-the-art photoelectrochemical performance and achieve high yield in C-H functionalization. This carbon nitride photoelectrode synthesis strategy may be readily adapted to other reported processes to optimize their performance.

Photoelectrochemistry (PEC) provides an environmentally benign and sustainable pathway for the direct conversion of solar energy into chemical fuels[1,2]. While PEC cells have been extensively studied for water splitting and carbon dioxide reduction[3,4], their potential for organic synthesis has been realized only recently[5]. Compared to photochemistry, the external potential in PEC guarantees fast charge separation, high atom economy, and high reaction efficiency[6,7]. In contrast to traditional electrochemistry, the lower applied voltage in PEC saves input energy and avoids excessive oxidation, which reduces side reactions[8]. Photoelectrodes are typically composed of inorganic semiconductor materials such as metal oxides (e.g., BiVO$_4$)[1,9–11] and III−V compounds (e.g., GaAs)[12,13], since they are stable under strongly oxidizing conditions and relatively easy to synthesize[14]. These metal-

based semiconductor materials suffer from inherent thermodynamic instability in a wide pH range at relevant external potentials[12], hindering their practical application.

Enabled by the versatility of synthetic chemistry, organic semiconductors with flexibly tunable properties have been increasingly used as active materials in diverse photoelectric devices[15]. One of the most appealing organic semiconductors for PEC is graphitic carbon nitride (CN)[16,17]. The reported CN films exhibited a ~55 times higher mobility than BiVO$_4$[18,19], and a much longer lifetime than BiVO$_4$ and WO$_3$[18,20] due to a triplet excited state[21,22]. Nevertheless, the development of organic semiconductor-based photoelectrodes is still in its infancy. Great efforts have been devoted to the design of the composition, including elemental doping[23–26] or formation of

[1]Max Planck Institute of Colloids and Interfaces, Am Muehlenberg 1, 14476 Potsdam, Germany. [2]Department of Chemistry and Biochemistry, Freie Universität Berlin, Arnimallee 22, 14195 Berlin, Germany. [3]Department of Materials, Department of Bioengineering, and Institute of Biomedical Engineering, Imperial College London, London SW7 2AZ, UK. [4]Institute for Applied Materials (IAM) and Karlsruhe Nano Micro Facility (KNMFi), Karlsruhe Institute of Technology (KIT), Hermann-von-Helmholtz-Platz 1, 76344 Eggenstein-Leopoldshafen, Germany. [5]Department of Chemistry, The Chinese University of Hong Kong, Shatin, New Territories, Hong Kong, China. ✉e-mail: Oleksandr.Savatieiev@mpikg.mpg.de; Felix.Loeffler@mpikg.mpg.de

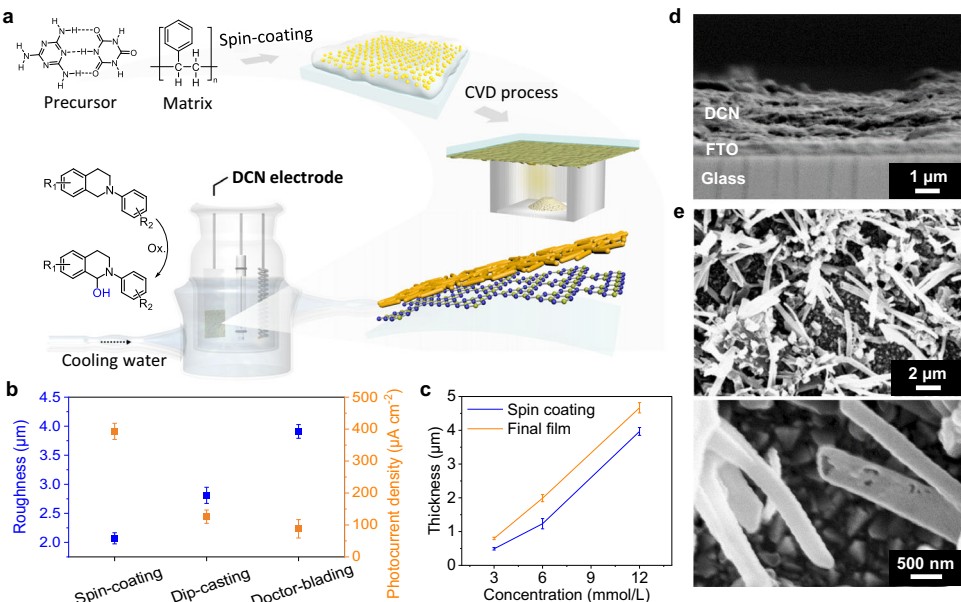

**Fig. 1 | Photoanode synthesis and dual-layer structure. a** Synthesis of dual-layer carbon nitride (DCN) by CVD with spin-coating pretreatment. **b** Comparison of different film-forming methods. Surface roughness is calculated by Root-mean square height. Photocurrent density is measured in 0.1 M Na$_2$SO$_4$ at 1.23 eV vs. RHE. Values are given as average ($n$ = 3) with standard deviation. **c** Relationship between film thickness and concentration of polymer matrix during pretreatment. Values are given as average ($n$ = 3) with standard deviation. **d** Cross-section and (**e**) surface view SEM image of DCN electrode (compared to standard CN, Fig. S5). Source data are provided as a source data file.

heterojunctions with other semiconductors[26,27], which modestly improved the properties of CN-based photoelectrodes (Fig. 1). PEC performance depends heavily on the quality of the film. Doctor blading, mainly employed today[28–30], is insufficient to generate fine films on conductive substrates with high affinity while chemical vapor deposition (CVD)[31,32] and vapor-deposition polymerization[33–36] provide high-quality films on various surfaces. While purely inorganic reactions, such as water splitting and carbon dioxide reduction, have been extensively studied with CN-based electrodes, for organic reactions, they are currently limited to straightforward PEC alcohol oxidations in aqueous electrolytes[37,38]. C–H functionalization using high-quality CN-based photoelectrodes would be very desirable.

Here, we report a dual-layer carbon nitride (DCN) electrode with photocurrent densities of up to 910 μA cm$^{-2}$ at 1.23 V vs. reversible hydrogen electrode (RHE). The DCN electrodes are prepared by spin-coating pretreatment, followed by co-condensation in a polymer film reactor during CVD. Spin-coating guarantees for a homogenously distributed precursor film with low roughness. The polymer matrix offers a carbon source for co-condensation to improve the conductivity at the molecular level. The pressure caused by the gas vapor in a sealed crucible, together with the precursor layer, enables the formation of a porous microtubular top layer and an interlayer between the porous film and the fluorine-doped tin oxide (FTO) substrate, resulting in high affinity. Our synthesis method not only produces a high-performance metal-free photoelectrode, but also allows for the facile optimization of existing electrode preparation methods. Higher photocurrents can be achieved by simply adopting spin-coating and the CVD setup to the reported protocols. Finally, we applied our DCN photoelectrodes in organic electrolytes for C–H functionalization, resulting in significantly higher yield than electrochemical approaches using standard platinum (Pt) working electrodes or photocatalytic approaches.

## Results

### Photoanode synthesis and dual-layer structure

The DCN photoanodes were synthesized by introducing a polymer matrix during the spin-coating pretreatment. Melamine-cyanuric acid (MCA) supramolecules were dispersed in a solvent together with a dissolved polymer. The resulting solution was spin-coated onto FTO glass to form a homogeneous film (Fig. 1a). The choice of solvent depends on the solubility of the polymer, for example, dichloromethane for polystyrene (PS) and water for polyvinyl alcohol (PVA). Although many CN precursors are typically insoluble in common solvents, we can prepare a uniformly distributed film with the polymer matrix acting as a film-forming agent (Fig. S1). The precursor film on the FTO glass is then subjected to a CVD process to obtain the final electrodes. To investigate the influence of different film generation methods, we prepared the precursor films with the same formulation (50 mg MCA, 20 mg PS, 500 μL DCM), but using different approaches: spin-coating, doctor blading, or dip-casting. By spin-coating, the prepared precursor films show the lowest surface roughness and the final electrodes exhibit the highest photocurrent densities (Figs. 1b and S2). These results suggest that spin-coating is more suitable to provide a fine film than doctor-blading and dip-casting, which contributes to a better PEC performance. Since the polymers we use are not cross-linked, their solution generally appears as a non-Newtonian liquid. During spin-coating, low viscosity caused by shear thinning can push the solution to quickly cover the substrate. Meanwhile, rotation of the substrate causes rapid evaporation of the solvents to reduce potential dewetting. Spin-coating not only produces more homogenous films than doctor-blading and dip-casting (Fig. S2), but also extends the scope of conductive substrate materials (Fig. S3) that can be used for the design of high-performance photoelectrodes.

The total thickness of the DCN film is about 1.8 μm, as shown by cross-sectional SEM and vertical scanning interferometry (Figs. 1d and S4e). Different film thicknesses can be obtained by varying the concentration of the polymer matrix (Figs. 1c and S4). Moreover, the final film is always slightly thicker than the spin-coated film (Fig. 1c), indicating that the main constituent layer is derived from the film in the pretreatment step. SEM images (Fig. 1d) clearly show a porous top layer with microtubular structures. Interestingly, a thin interlayer was observed between the top layer and the substrate. After scratching off the top porous layer, this interlayer appears as a transparent and condensed yellowish film with a thickness of

around 100 nm (Fig. S5). If there is no precursor (melamine) in the crucible during the CVD process, no interlayer is formed. Therefore, this interlayer is formed by the precursor vapor during the CVD step. Unlike the typical CVD setup where the substrate is placed in a tilted test tube, we place the substrate on top of a crucible with the pretreatment supramolecular layer facing the inside of the crucible, forming an enclosure with increased vapor pressure. In this case, the vapor not only passes over the supramolecular layer, but can also penetrate the layer at high temperatures. The porous top layer increases the surface area of the DCN electrodes and the bottom interlayer improves the affinity, which affects the charge transport capability within the electrode material.

## Synthesis mechanism

We deposited a carbon nitride thin film on an FTO substrate without the spin-coating step to compare its performance with our DCN electrodes. In the infrared spectrum (Fig. S6), the DCN shows similar characteristics to the CN films, indicating good retention of the heptazine rings. The main difference is the decreased intensity of the peak between 3000 and 3500 cm$^{-1}$, which is attributed to N-H stretching vibration, suggesting a higher polymerization degree in DCN. During the CVD process, the state of the polymer matrix gradually changes as temperature increases. It becomes viscous above the glass transition temperature of 67.2 °C[39], allowing diffusion similar to a solvent layer[40,41], followed by its decomposition at about 400 °C (Fig. S5), and the formation of a carbon skeleton (compare Fig. 2d). When the temperature exceeds 500 °C, the carbon skeleton gradually disappears. Therefore, the polymer plays different roles during DCN synthesis. To further investigate the role of the polymer matrix during the electrode synthesis, we analyzed its surface by X-ray photoelectron spectroscopy (XPS). The peaks at 293.7 eV (π-π*), 288.5 eV (N-C = N), and 286.9 eV (CO/C-pyrrolic) in the C 1$s$ spectrum originate from the heptazine rings, while the peak at 285 eV suggests the presence of C-C/C–H bonds (Fig. 2a). To exclude the influence of surface contamination, XPS analyses were performed at different times of Argon cluster ion etching (Fig. 2b). Notably, even after 5 h of surface etching, the peak was still present (Fig. 2c), indicating that the C-C bonds are indeed part of the DCN film structure introduced during the process. Thus, the polymer serves three purposes at different stages of the synthesis process: It is a film-forming agent during spin-coating (<25 °C), a reaction confinement for CN

formation (>67 °C), and a carbon backbone during carbonization (>400 °C).

Next, we analyzed the C/N ratio of the photoelectrodes at different temperatures (Fig. 2d). As a reference (DCN$_{MCA}$), the polymer was omitted during spin-coating. The C/N ratio of DCN$_{MCA}$ gradually increases at higher temperatures due to the elimination of amino groups during condensation. The DCN electrodes generally show a higher C/N ratio than DCN$_{MCA}$ due to the contribution of the polymer. When the temperature exceeds 500 °C, the C/N ratio of DCN decreases dramatically, indicating that the polymer begins to decompose. While in thermogravimetric analysis the pure polymer starts to decompose already at 380 °C, an interaction between the precursor and the polymer could explain the upshift of the polymer decomposition and the observed C-C bonds in the DCN electrodes at elevated temperatures. The C/N ratio in the heptazine rings stabilizes at 0.73 ± 0.03 after surface cleaning, which is quite close to the theoretical maximum for perfectly polymerized carbon nitride. In fact, the polymer contributes to a better condensation process (Fig. 2f). The polymerization degree of the melem groups was quantified by $^{15}$N solid-state NMR analysis. After deconvolution, the ratio between the integral intensities of I(NH$_2$) and I(NH) was calculated to be 0.39 and 0.55 for DCN and CN, respectively (Fig. 2e & Fig. S7). These oligomeric units of DCN and CN may contain ten and six melem subunits, respectively, since the spin-coating pretreatment contributes to the formation of larger oligomeric units with a higher polymerization degree.

## Parameter optimization and PEC properties characterization

DCN electrodes synthesized under different conditions show different current densities (Fig. 3a). Four polymer matrices were investigated: PS, polystyrene-acrylic copolymer (S-LEC), polyvinylpyrrolidone (PVP), and polyethylene glycol (PEG). Both PS and S-LEC show high performance at slightly higher synthesis temperatures (560–570 °C), while PEG requires relatively lower temperatures (550–560 °C). In the case of PVP, the FTO substrates are burnt and the electrodes are over-carbonized. Therefore, PVP is not suitable for this synthesis method. In aqueous electrolyte, photoanodes prepared with 60% w/w PS at 560 °C gave the highest current density of 910 µA cm$^{-2}$ (~700 ± 200 µA cm$^{-2}$ in average of two batches of photoelectrodes) and were further characterized. To date, the main strategy to improve the performance of CN electrodes has been to form heterojunctions with noble metals or transition metal oxides. In contrast, DCN

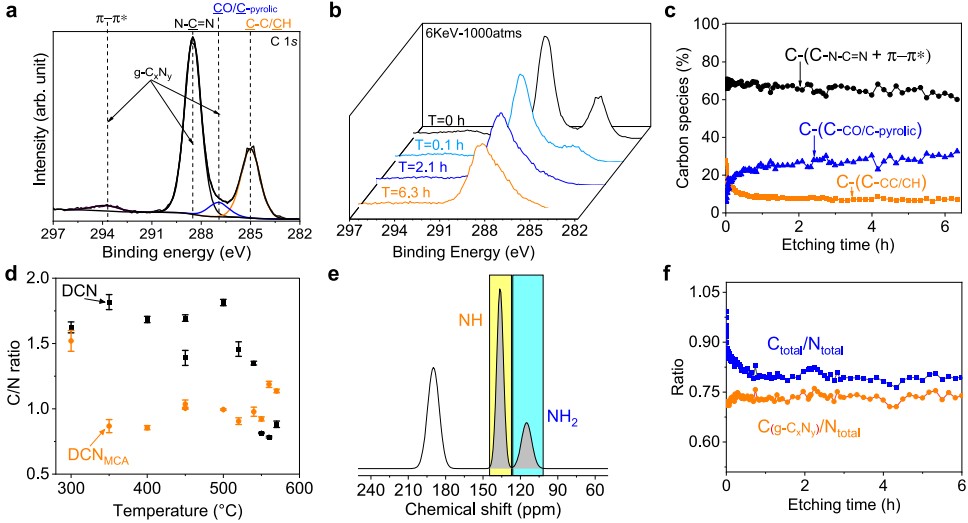

**Fig. 2 | Mechanism analysis of the synthesis process. a** High-resolution XPS C 1$s$ spectrum of DCN photoelectrodes. **b** XPS C 1$s$ spectra under Argon cluster ion etching (6 keV Ar$_{1000}$$^{+}$). **c** XPS depth profile of the carbon species ratio. **d** C/N ratio of photoelectrodes at different temperatures. Data plots are measured by energy dispersive X-Ray analysis. Values are given as average ($n$ = 3) with standard deviation. **e** $^{15}$N solid-state NMR spectrum of DCN films after deconvolution (Bruker 600, 8 kHz, CP/MAS). **f** XPS depth profile of the carbon/nitrogen ratios. Source data are provided as a source data file.

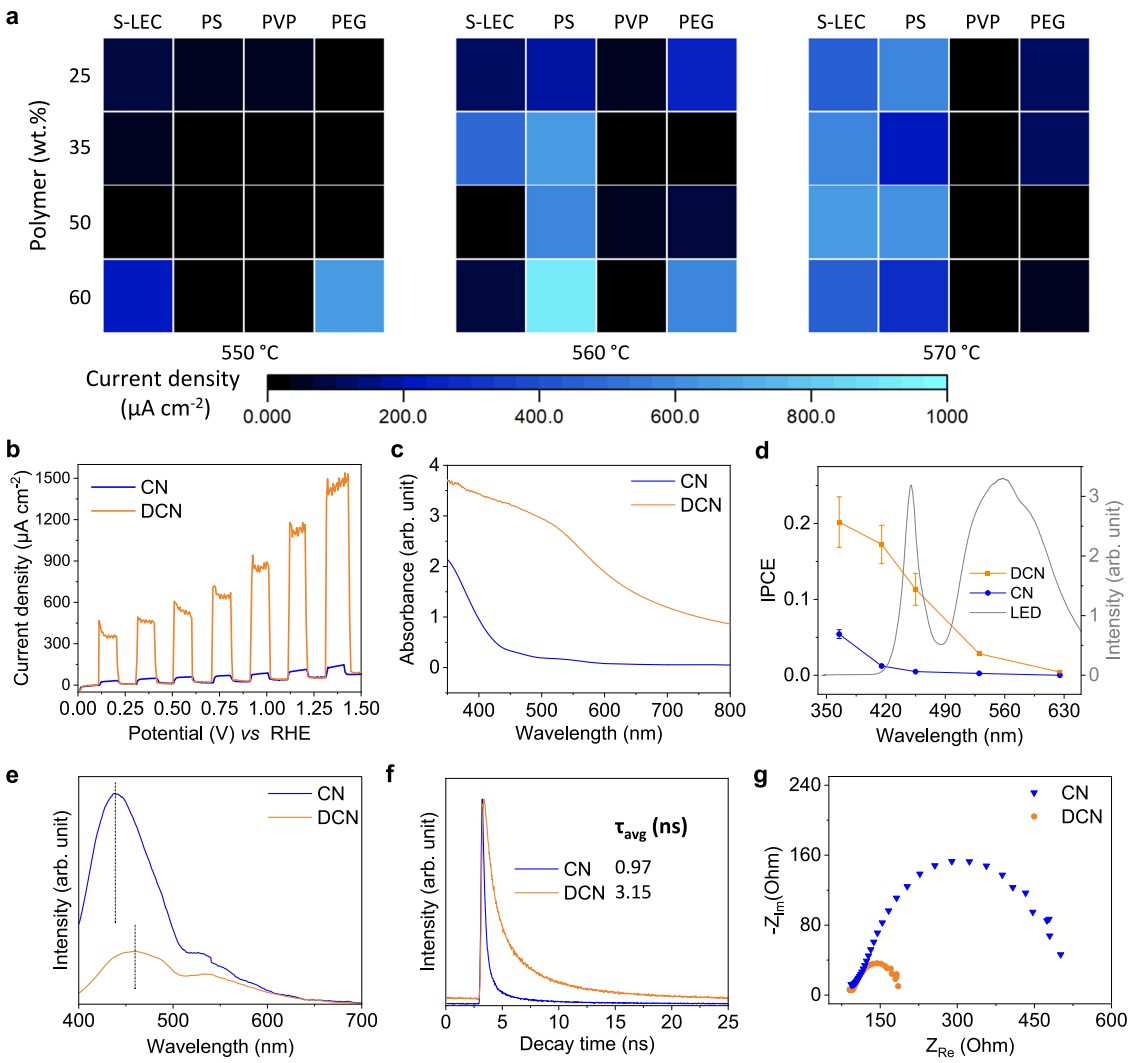

**Fig. 3 | PEC performance of optimized photoanodes in water. a** Current density of DCN electrodes under different synthesis conditions: polymer type and concentration, temperature, at +1.23 V vs. RHE with 0.1 M Na₂SO₄ in water (see Table S2). These values were obtained by subtracting the dark current density from that measured under light irradiation. **b** Representative linear sweep voltammetry (LSV) curves of DCN and CN electrodes in 0.1 M Na₂SO₄ with and without light. **c** UV-visible absorption spectra. **d** Incident photon-to-electron conversion efficiency (IPCE) plots and white light LED spectrum used for PEC reactions. Values are given as average (*n* = 5) with standard deviation. **e** Steady-state photoluminescence (PL) emission spectra acquired upon excitation at 365 nm. **f** Time-resolved PL decay spectra at 530 nm acquired upon excitation at 375 nm. **g** Electrochemical impedance spectroscopy (EIS) Nyquist plots of electrodes in the dark. Source data are provided as a source data file.

photoelectrodes exhibit much higher PEC properties than most reported CN-based electrodes, even without metals. Contamination of DCN with trace metals could be ruled out by inductively coupled plasma optical emission spectroscopy (Table S1 and Supplementary Discussion 3, most notably -0.02 % of Cu, Zn, K, Ca).

Linear sweep voltammetry of the electrodes (Fig. 3b) shows about 100 times higher current density and reduced onset of oxidation current under white light irradiation for DCN. A redshift of up to 800 nm is observed in the UV-vis spectrum (Fig. 3c). Since it is unlikely to be caused by scattering alone (Fig. S8), it suggests that dopant and defect-related midgap states (MS) are generated in the forbidden gap of normal CN. The calculated incident photon-to-current conversion efficiency (IPCE) of DCN electrodes at different illumination wavelengths (Fig. 3d) shows a direct correlation with light absorption. A high IPCE value of 20.2% is achieved at 365 nm, and a photoresponse can be detected up to 625 nm. In addition, significant photoluminescence (PL) quenching occurs on the DCN electrodes in the steady-state PL emission spectra (Fig. 3e), indicating strongly inhibited recombination of photogenerated electron-hole pairs. The redshift of

the main PL peak in DCN indicates charge migration through the interface of the dual-layer structure. The time-resolved transient PL decay spectra further demonstrate that DCN has a longer average charge carrier lifetime ($\tau_{avg}$) compared to CN electrodes (Fig. 3e). The steady-state fluorescence spectrum of DCN was acquired upon excitation at 375 nm (Fig. S9). The DCN electrodes show a smaller semicircle compared to the CN electrodes (Fig. 3g), indicating a higher charge mobility for the DCN electrodes. Furthermore, the bandgaps of the photoelectrodes are calculated to be 1.8 eV for the top layer and 2.7 eV for the bottom layer by the transformed Kubelka-Munk function (Fig. S5). Based on Mott-Schottky (MS) plots, which provide information on the flat band potential of semiconductor materials (Fig. S10), we estimated the conduction band edges of the layers to be −0.77 V vs. RHE and −0.96 V vs. RHE for the top and bottom layers respectively. The narrow bandgap of the top layer can harvest a large portion of the solar spectrum. The valence band energy of the two layers differs significantly, resulting in a directed flow of photogenerated holes into the top layer. These holes can further contribute to the substrate oxidation at the anode.

## Performance of photoanodes in non-aqueous electrolytes

The high PEC properties make DCN photoanodes interesting for organic transformations. The oxidation of *N*-aryl-tetrahydroisoquinolines (THIQ), commonly achieved by photocatalytic transformation, was selected to test the performance of our DCN photoelectrodes. Due to its stable redox properties in different solvents, the ferrocenium/ferrocene couple, Fc$^+$/Fc, is commonly used as an internal standard for organic systems (Fig. S11). According to cyclic voltammetry (Fig. 4a), the half peak oxidation potential, $E_{p1/2}$, of THIQ is about +0.22 V vs. Fc$^+$/Fc. To optimize the electrodes for organic solvents, photocurrent densities of DCN electrodes were measured in methanol under the oxidation potential of THIQ (Fig. 4b). Photoanodes generated with 50 % w/w PS at 560 °C give the highest current densities of 578 µA/cm$^2$, which were selected for the subsequent organic transformations. Their transient photocurrent density under interval light irradiation shows a fast light response and stable current density (Fig. 4c) with sufficient long-term stability (Fig. S12).

## Reaction optimization and scope evaluation

Based on the results (Fig. 4), the potential was set to +0.22 V vs. Fc$^+$/Fc, resulting in a photocurrent density of ~550–600 µA cm$^{-2}$ using 0.1 M LiClO$_4$ in methanol. The reaction time was initially set to 5 h and tested with several organic solvents (Table 1, entries 1–4). The crude products were analyzed by NMR spectroscopy without further purification. Yields were calculated using CH$_2$Br$_2$ as an internal standard. Quantitative conversion of the starting material was achieved within 2 h in methanol (Table 1, entry 5). A Pt mesh anode was used in the control experiment (Table 1, entry 6). Since the Pt electrode is not responsive to light irradiation, a potential of +0.52 V vs. Fc$^+$/Fc is necessary to reach the same current density. The yield reached only 19 % after 2 h, suggesting that the presence of light irradiation can

not only reduce the input of electricity, but also significantly improve the reaction efficiency. The lower efficiency of the reaction using a Pt anode is in agreement with the previously reported electrochemical functionalization of N-phenyl-tetrahydroisoquinolines at the benzylic position next to the nitrogen atom, where the carbon anode was found to be more effective compared to Pt[42]. Another control experiment was performed without an applied (electrical) bias (Table 1, entry 7). In this situation, the DCN electrode can only act as a photocatalyst and no product was detected after 2 h. The PEC approach provides a highly efficient and environmentally friendly means of THIQ oxidation. With the optimized conditions (Table 1, entry 5), the scope of photoelectrocatalytic C−H functionalization was investigated (Table 2). First, monosubstituted substrates were tested. The oxidation of monomethoxylated THIQ derivatives gives high yields (2b, 2c). Multisubstituted THIQs (1d, 1e) were converted into the corresponding products. The yield of 2e was relatively lower, possibly due to the formation of side products through overoxidation. In addition, THIQ with an electron-withdrawing trifluoromethyl group (1f) was also oxygenated under these PEC conditions with acceptable product yields. Considering published data[42,43], we outlined the photoelectrochemical hydroxylation mechanism of *N*-aryl-tetrahydroisoquinolines (Fig. S13 and Supplementary Discussion 1). Considering a photocurrent density of 500 µA cm$^{-2}$ with 1 cm$^2$ of the photoelectrode immersed in the electrolyte, a 2 h duration of the photoelectrocatalytic experiment, a 50 µmol loading of 1a-f, and the yields for 2a-f, the Faradaic efficiency was determined to be ~67% (2a-d), ~60% (2e), and ~33% (2 f).

To demonstrate the potential of the DCN electrodes for other cross-coupling reactions, we investigated a proof-of-concept synthesis of 4-(4-methylphenyl)-morpholine (Fig. S14 and Supplementary Discussion 2).

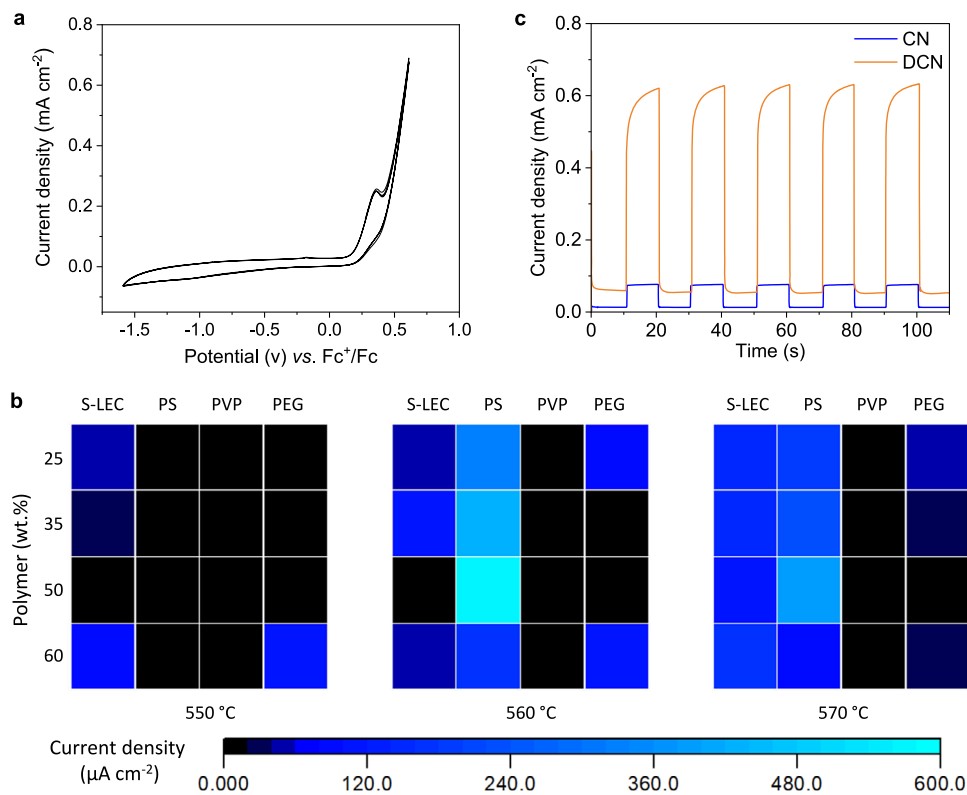

**Fig. 4 | Performance of photoanodes in non-aqueous electrolyte. a** Cyclic voltammogram of THIQ with platinum mesh as working electrode. **b** Current density of DCN electrodes under different synthesis conditions: polymer type and concentration, and temperature, at +0.22 V vs. Fc$^+$/Fc (see Table S3). **c** Transient photocurrent density response of DCN and CN electrodes at +0.22 V vs. Fc$^+$/Fc. All experiments were performed using 0.1 M LiClO$_4$ in methanol. Source data are provided as a source data file.

**Table 1 | Optimization of reaction conditions for the photoelectrocatalytic C–H functionalization with 50 µmol of substrate in 5 ml solvent and 0.1 M LiClO4 as electrolyte**

| Entry | Anode | Applied bias[a] | Reaction time | Light | Solvent | Yield[b] |
|---|---|---|---|---|---|---|
| 1 | DCN | 0.22 V | 5 h | White light | ACN | 86% |
| 2 | DCN | 0.22 V | 5 h | White light | Acetone | 86% |
| 3 | DCN | 0.22 V | 5 h | White light | DMSO | Quant. |
| 4 | DCN | 0.22 V | 5 h | White light | MeOH | Quant. |
| 5 | DCN | 0.22 V | 2 h | White light | MeOH | Quant. |
| 6 | Pt mesh | 0.52 V | 2 h | No light | MeOH | 19% |
| 7 | DCN | No bias | 2 h | White light | MeOH | 0 |

[a]Applied bias vs. Fc⁺/Fc electrode.
[b]Yield is determined by NMR spectrum with $CH_2Br_2$ as the internal standard.

## Discussion

We have developed a facile method for the synthesis of DCN photoanodes with high PEC performance. The introduction of a spin coating step with a multipurpose polymer results in the dual-layer structure of the electrodes: the porous top layer increases the interfacial area with the electrolyte, and the thin bottom layer improves the affinity between the films and the substrate. The dopant and defect-related midgap states (MS) in the forbidden band of normal CN contribute to efficient charge migration and separation, resulting in long charge lifetimes. Since spin coating generally provides better defined films than doctor blading, the currently dominant method for CN-based electrode preparation, our approach can be readily adopted to optimize many other processes to improve photoelectrode fabrication. In addition, we show that metal-free photoanodes can be used for C–H functionalization. A quantitative yield for the oxidation of THIQ can be achieved without redox mediators, which is not possible with commercially available platinum mesh electrodes, even at higher applied potentials. For future scale-up, our method could be transferred to a blade coating or similar process, but will require optimization. Therefore, this work opens a new avenue for cost-effective and environmentally friendly metal-free photoanodes in organic synthesis.

## Methods

### Electrode preparation

In all, 100 mg MCA precursor and 100 mg (co)polymer (polystyrene (PS) 35k, Sigma-Aldrich; styrene acrylic copolymer, S-LEC PLT 7552, Sekisui Chemical GmbH, Germany, polyvinylpyrrolidone (PVP), av. Mw = 2000–3000, Sigma-Aldrich; polyvinyl alcohol (PVA), av. Mw = 9000–10,000, Sigma-Aldrich; polyethylene glycol (PEG), av. Mw ≈ 20,000, Sigma-Aldrich) were dissolved in 1 mL dichloromethane. For the preparation of MCA supramolecules, 1.29 g cyanuric acid was mixed with 1.26 g melamine in 50 mL water. Then, the mixture was shaken overnight and collected by centrifugation. All solvents were used as obtained from Sigma-Aldrich. We spin-coated the solution on a clean fluorine-doped tin oxide (FTO) coated glass (6 cm × 6 cm, 7 Ω/sq, Merck). This pre-treated FTO was used to cover a 29.5 mL rectangular alumina crucible with 5 g melamine (99%, Alfa Aesar). We placed the crucible in a furnace (CMF-1200, Carbolite Gero, UK), set it to ~560 °C, incrementing by 1.5 °C/min and kept it at 560 °C for 3 h under nitrogen.

The as-prepared DCN slides were cut into small pieces (1 × 1 cm² on 1 × 1.5 cm² FTO) and used as photoelectrodes. The volume of the carbon nitride layer on the FTO was about 1 cm × 1 cm × 1.8 µm (L × W × H) with a gravimetric density of ~2 g cm⁻³. The mass loading of carbon nitride on a standard photoelectrode was ~0.36 mg. Except for sodium, metal contents were around or, in most cases, much lower than 0.005 mol % (Table S1), which can be considered as trace amounts in a standard carbon nitride photoelectrode. For standard CN electrodes, the FTO glass on top of the crucibles was not pre-treated.

### Analytical methods

We used a white light interferometer smartWLI compact (Gesellschaft für Bild- und Signalverarbeitung (GBS) mbH, Illmenau, Germany) with low magnification (5x Nikon CF IC Epi Plan DI - Mirau) to acquire the thickness maps and profiles in large area.

Steady-state photoluminescence (PL) emission spectra were acquired with a JASCO FP-8300 fluorescence spectrometer upon excitation at 365 nm.

The time-resolved transient PL decay spectra were measured at room temperature using a TCSPC FluoTime 250 (PicoQuant) spectrometer equipped with a PDL 800-D picosecond pulsed diode laser driver. Equations (1) and (2) were applied to carry out the di-exponential kinetic analysis and to calculate the average lifetime $\tau_{avg}$, respectively.

$$y = \sum a_i e^{-\left(\frac{t}{\tau_i}\right)} \qquad (1)$$

$$\bar{\tau} = \frac{\sum a_i \tau_i^2}{\sum a_i \tau_i} \qquad (2)$$

where $y$ is the intensity usually assumed to decay as the sum of individual exponential decays, $a_i$ is the pre-exponential factor and $\tau_i$ is the decay time.

UV-vis absorption measurements were performed in transmittance mode using a PG Instruments TG70 + UV/vis spectrometer.

Incident photon-to-current conversion efficiency (IPCE) measurements were carried out at +1.23 V vs. RHE and IPCE was calculated by the following formula:

$$IPCE(\%) = \frac{J_{Ph}(A/cm^2) \times 1240(V \cdot nm)}{\lambda(nm) \times J_{Light}(W/cm^2)} \qquad (3)$$

where $J_{Ph}$ is the photocurrent density, $\lambda$ the wavelength, and $J_{Light}$ is the intensity of incident light. The flat band potential was calculated with

**Table 2 | Scope of N-aryl-tetrahydroisoquinoline derivatives for the photoelectrocatalytic C–H functionalization**

DCN photoanode
0.22 V vs. Fc⁺/Fc, 2 h
Pt cathode, undivided cell
0.1 M LiClO₄ in MeOH
White LED, 100 mW/cm²

**2a**, 100%   **2b**, 100%   **2c**, 100%   **2d**, 100%

**2e**, 86%   **2f**, 49%

the Mott-Schottky equation (15 Hz, without light):

$$\frac{1}{C^2} = \frac{2}{N_D e \varepsilon \varepsilon_0} \cdot \left[ \left( V_S - V_{fb} \right) - \frac{k_B T}{e} \right] \quad (4)$$

where $C$ is the space-charge capacitance, $V_S$ is the applied potential, $V_{fb}$ is the flat band potential, $N_D$ is the electron carrier density, $\varepsilon$ is the relative permittivity of the semiconductor, $\varepsilon_O$ is the permittivity of the vacuum, $e$ is the elementary charge, and $k_B$ is the Boltzmann constant.

### Synthesis of *N*-aryl-tetrahydroisoquinoline derivatives

*N*-aryl-tetrahydroisoquinoline derivatives (THIQs) were synthesized as reported[44]. Briefly, for N-aryl-tetrahydroisoquinoline, copper(I)iodide (190.5 mg, 1.0 mmol) and potassium phosphate (4245.4 mg, 20.0 mmol) were placed into a flask. They were evacuated and flushed with argon. 2-Propanol (20.0 mL), ethylene glycol (1241.4 mL, 20.0 mmol), 1,2,3,4-tetrahydroisoquinoline (1331.9 mg, 10.0 mmol), and iodobenzene (2.1 mL, 15.0 mmol) were added via syringes at room temperature. The mixture was heated to 85 °C for 24 h and then cooled to room temperature. Then, ethyl acetate (20 mL) and water (40 mL) were added. The aqueous layer was extracted by ethyl acetate (2×50 mL). The combined organic phases were washed with brine and dried over magnesium sulfate. After removing the solvent, the crude product was purified by column chromatography.

### Photoelectrochemical reaction/oxidation

PEC reactions were performed with a three-electrode potentiostat (BioLogic MPG2) with a DCN or CN photoanode (for PEC experiments) or Pt mesh (electrochemistry experiments) as the working electrode and a Pt mesh as the counter electrode (CE). Pt mesh electrodes were acquired from ALS Electrochemistry & Spectroelectrochemistry Inc (Catalog No. 012961) and used without further treatment or modifications. Ag/AgNO₃ or Ag/AgCl reference electrodes (RE) were used for organic and aqueous solutions, respectively, without iR-compensation. All photoelectrochemical reactions were carried out in a chamber with a cooling system, which ensured a constant temperature of 20 °C. No electrode pre-activation was involved in this project.

For PEC reactions, 106.4 mg LiClO₄ was dissolved in 5 ml of solvent (typically methanol). Then, 50 μmol of the THIQ derivative was added to the electrolyte solution and flushed with nitrogen for 10 min. The reaction vessel (Fig. S15) was irradiated by a white LED with a density of 100 mW/cm² for typically 2 h. To avoid high temperatures caused by light irradiation, a water-cooling jacket around the reaction vessel was needed. After the reaction, the solvent was removed via rotary evaporation at room temperature. The crude product was dissolved in 500 μl methanol-d₄ for NMR spectroscopy with CH₂Br₂ as the internal standard (example of crude vs. purified NMR, Fig. S16).

Photocurrents were measured after three minutes of equilibration time. The photocurrent with reference to reversible hydrogen electrode (RHE) is calculated by using the following formula:

$$E_{(RHE)} = E_{Ag/AgCl} + 0.059 \, pH + E^0_{Ag/AgCl} \quad (5)$$

where $E_{Ag/AgCl}$ is the applied working potential and $E^0_{Ag/AgCl} = 0.1976$ V at 25 °C.

### Faradaic efficiency

Faradaic efficiency measurements were performed at +0.22 V vs. Fc + /Fc electrode. The Faradaic efficiency (F.E.) was calculated according to the following equation:

$$\text{F.E.} = \frac{n_p}{n_e} \times 100\% = \frac{n_r \times \eta}{\left( \frac{N_e \times j \times S \times t}{q_e \times N_A} \right)} \quad (6)$$

with $n_p$ – amount of product generated in the photoelectrocatalytic experiment, mol; $n_e$ – amount of electrons passed through the PEC cell, mol; $n_r$ – loading of substrate, mol; $\eta$ – chemical yield of product, %; $N$ – number of electrons involved in the chemical transformation, i.e. $N_e = 2$ according to the proposed mechanism; $q_e$ – elementary charge, $1.6 \times 10^{-19}$ C; $j$ – average current density in the photoelectrochemical experiment, A cm⁻²; $S$ – area of the photoelectrode, cm²;

$t$ – time of the photoelectrochemical experiment, s; $N_A$ – Avogadro constant, $6.02 \times 10^{23} \, mol^{-1}$.

## Statistical analysis

Average photocurrent density ($<j>$) and the corresponding standard deviation ($s_N$) were calculated according to the equations:

$$<j> = \frac{\sum_{i=1}^{N} j_i}{N} \tag{7}$$

$$s_N = \sqrt{\frac{\sum_{i=1}^{N} (j_i - <j>)^2}{N}} \tag{8}$$

with $j_i$ – photocurrent density measured for the $i$th batch of the DCN photoelectrode, $\mu A \, cm^{-2}$. $N$ – the number of photoelectrodes batches prepared.

## Data availability

The authors declare that the main data supporting the findings of this study are available within the article and its Supplementary Information files or are available from the corresponding authors upon reasonable request. Source data are provided with this paper.

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

## Acknowledgements
This work was supported by the funding from the China Scholarship Council (J.Z.), the German Federal Ministry of Education and Research (BMBF, grant number 13XP5050A, F.F.L.), and the MPG-FhG cooperation (Glyco3Display, F.F.L.), and the Max Planck Society. J.Z. and M.M.S. acknowledge support from the grant from the UK Regenerative Medicine Platform Acellular/Smart Materials – 3D Architecture (MR/R015651/1). The authors thank Dr. Matthew Plutschack, Prof. Markus Antonietti, and Dr. Gaofeng Chen for fruitful discussions, as well as Olaf Niemeyer, Eva Settels, and Jessica Brandt for technical assistance. This work was partly carried out with the support of the Karlsruhe Nano Micro Facility (KNMFi, http://www.knmf.kit.edu), a Helmholtz Research Infrastructure at Karlsruhe Institute of Technology (KIT, http://www.kit.edu).

## Author contributions
J.Z. conceived the DCN material. J.Z., O.S., and F.F.L. designed and analyzed the experiments. J.Z. performed the experiments. Y.Z. supported with the THIQ synthesis and analysis. P.D. generally supported the project and performed NMR analyses. Y.L. supported the project and designed the figures. C.N. performed the XPS experiments and analyses. M.M.S. and P.H.S. generally supported the project, including funding support. O.S. designed some PEC reactions and supported the measurements and analysis. O.S. and F.F.L. supervised the project. The manuscript was mainly written by J.Z., as well as O.S. and F.F.L., while all authors contributed to the revision.

## Funding

## Competing interests
J.Z., Y.Z., O.S., and F.F.L. have filed a patent application for the DCN material (EP22151236.1). All other authors declare no competing interests.
