## [Peer Review File · Nature Communications]

Metal-free photoanodes for C-H functionalizationREVIEWER COMMENTS

Reviewer #1 (Remarks to the Author):

The present study addresses the synthesis of graphitic carbon nitride photoanodes for the oxidation of water and C-H functionalization. The authors have developed a new method to deposit the active semiconductor material, improving the performance with respect to the reference material. The study is interesting, but needs further work and there are many issues, which need to be clarified before the manuscript can be further considered for publication in Nature Communications. More detailed comments are provided below:

- 1) In the abstract, the expression "...suffers from film generation techniques." is ambiguous and does not clearly describe the motivation of the present study.
- 2) The authors claim that: "All CN-based electrodes are currently limited to straightforward PEC alcohol oxidations in aqueous electrolytes.", but this is not true, and they have even cited different studies focusing on PEC water splitting (e.g.: Nat Commun 11, 4701)
- 3) The authors also claim that: "Here, we report a dual-layer carbon nitride (DCN) electrode with unprecedentedly high photocurrents of 910 $\mu\text{A cm}^{-2}$ at 1.23 V vs reversible hydrogen electrode (RHE)." This is not correct, the photocurrent is the current under illumination minus the dark current, which is around 500 $\mu\text{A cm}^{-2}$, according to figure 3b.
- 4) The spin-coating method developed is not adequate for large scale photoelectrodes. The authors should comment on the limitations of this method and how to overcome them considering the scalability of the proposed approach.
- 5) The description of the synthesis provided in Figure S1 are not very clear. The steps are not clearly indicated, which leads to confusion about how the dual layer is formed.
- 6) The authors claim that: "The total thickness of the DCN film is around 1.8 μm , as shown by cross-section SEM (Fig. 1d)." But from this figure, this thickness is not clearly visible.
- 7) The two distinct layers (insert layer and tubular layer) claimed by the authors are not visible in Figure 1.
- 8) In Figure S4, one of the scale bars is not labelled.
- 9) The authors claim that: "...enabling diffusion similar to a solvent layer, followed by its decomposition at around 400 $^{\circ}\text{C}$, and the formation of a carbon skeleton (Fig. S5)." But again, this is not clear from Figure S5.
- 10) For the photoelectrochemical characterization, the authors should clearly indicate the geometrical area of the tested films, information about performance statistics and why a Ag/AgNO₃ reference electrode was employed for aqueous solution, since this is not the most adequate reference electrode in these conditions (Figure 3).
- 11) The authors claim that: "A redshift up to 800
- 12) 195 nm is observed in the UV-vis spectrum (Fig. 3c), indicating that dopant and defect-related midgap states (MS) are generated in the forbidden gap of the normal CN." But this can also arise from diffuse scattering, there is not information at all on how the optical measurements were carried out.
- 13) The IPCE measurements should also be integrated with the solar spectrum to validate the photocurrent measurements reported in Figure 3a and b. There is no information on the specific potential at which IPCE measurements were carried out.
- 14) The authors discuss about Mott-Schottky plots, but this information is not in the manuscript or SI.
- 15) Additionally, these results from Figure 3 should be validated with H₂ and O₂ gas chromatography measurements to validate that the photoelectrodes are properly working.
- 16) The tests for organic transformation were carried out in methanol. Methanol is well-known as a hole scavenger in photoelectrochemical measurements, the authors should validate that this substrate was inert during the measurement.
- 17) Furthermore, the cyclic voltammetry curve showed in Figure 4a does not seem to be related to photocurrent and it is difficult to find the correlation with Figure 4b.
- 18) In Figure 4b, the measurement should be longer (not only 1 min) to validate that the photoelectrode is relatively stable under operation conditions.
- 19) The authors carried out a comparison of the organic transformation, using Pt as a reference. But,

how good is Pt to carry out this organic transformation? The authors should provide some benchmark to clearly understand the improvement with respect to the state of the art.

20) There are some typos to be corrected, for example there is no figure 5 in the manuscript, and References 26 and 27 are duplicated.

Reviewer #2 (Remarks to the Author):

Photoelectrochemistry (PEC) is gaining momentum in recent years by providing an environmentally benign and sustainable pathway to access (bio)molecules under mild reaction conditions. The work "Metal-free photoanodes for C-H functionalization" by Savateev, Loeffler, and co-workers is a comprehensive study for the synthesis of dual-layer CN photoelectrodes with improved photoelectrochemical performance. This is true that the C-H functionalization or any other C-C OR C-N bond-forming reaction employing such photoelectrodes (PEC in general) is highly desirable from the viewpoint of sustainable chemistry and would provide an initial benchmark for future development in this field.

The CN photoelectrodes described in this work exhibit improved PEC performance with quantitative yields for most of the substrates for the said C-H functionalization with just an acceptable substrate scope. As I would have preferred an extended scope for this reaction (e.g. Dimethylanilines) to see the limitation of such a strategy. If authors have already faced such issues it should be a valuable addition to the SI of this MS. On the other hand, I am a bit skeptical about the role of trace impurities (0.02% of Cu, Zn, K, Ca) in such dual-layer CN electrodes as Cu is known to cause similar oxidations. Overall, this method has the potential to provide a general platform for metal-free C-H oxidations employing photo-electro-catalysis. The method will make a nice addition to the ever-growing field of synthetic photoelectrochemistry.

Specific comments:

- a) The scale of the optimization reaction should be mentioned in the footnote of Table 1. Did authors screen other electrodes (cathode) than Pt as Pt is an expensive metal for such oxidations?
- b) Also the details of experimental conditions whether in MS or SI seems to be missing. I would also prefer to see details of the experimental setup for the benefit of readers.
- c) The SI should report complete synthetic procedures and characterization of the final products according to the journal guidelines. To me it is not clear whether these are crude NMRs or purified ones so maybe a detail in the SI would be helpful. Also, the NMR spectra for most of the compounds are not pure which eventually also affects the isolated yield of the reaction. So please comment. If all are crude at least one example should be confirmed via isolated yield.

Reviewer #3 (Remarks to the Author):

In this paper, the authors report the fabrication of CN film by using a new deposition method. The authors use the as-prepared CN electrode for C-H functionalization. The work is interesting as they introduce a new approach for fabricating new CN electrodes and the photoanode measurements in an organic medium. I believe this work can be published in Nature Communications, but the authors should carefully revise it.

1. I would like to kindly recommend the authors avoid writing extraordinary PEC performance, unprecedentedly high photocurrents, etc. It doesn't contribute to the discussion, and in some cases, it is misleading.
2. The LSV data (Fig 3b) should be obtained using an electrolyte solution purged with N₂/Ar.
3. the authors incorrectly mentioned a photocurrent density of 910 A/cm² in their study. The authors should subtract the dark current value to obtain the accurate photocurrent density. I recommend

conducting chronoamperometry measurements (light on/off) of the electrodes and providing the corresponding data. In addition, statements such as: "unprecedentedly high photocurrents of 910 $\mu\text{A cm}^{-2}$ at 1.23 V vs. reversible hydrogen electrode" are slightly too misleading as there are reports with similar current densities. In addition, the photocurrent has no meaning. The Faradic efficiency and stability are more important. Does the current due to water oxidation? Or is it due to self-oxidation?

4. It is essential for the authors to conduct a stability test on the electrodes, as the stability directly impacts the efficiency of the electrodes for practical applications. Furthermore, post-characterization should be performed after the stability test.

5. In Figure 3f, the authors measured the lifetime at 530 nm, while the emission maximum is between 450-460 nm (Fig. 3e). This discrepancy should be explained.

6. The authors claimed that the activity of the DCN sample is due to the formation of C-C bonds. Consequently, these bonds should be visible in the FTIR spectrum of DCN (Fig. S6). Additionally, they should be observed in the ^{13}C solid-state NMR of DCN (Fig. 2e, Fig. S7).

7. The authors presented three methods for constructing a CN film: doctor-blading, spin coating, and dip-coating, using DCM as the solvent. However, while DCM might be a suitable solvent for spin coating, it may not be suitable for other methods, such as doctor blading. Therefore, I recommend the authors explore alternative solvents such as ethylene glycol, and for fair comparison (which I don't think is needed), they should adopt the best method from the literature.

8. The authors should also consider moderating their statement in the abstract: "...but their photoelectrochemical (PEC) performance suffers from film generation techniques.". In the last years, many methods have been introduced to develop excellent CN films. The PEC performance may suffer from self-oxidation of CN, inefficient charge separation, etc.

9. Mechanistic investigation for the C-H functionalization using DCN film is missing. The charge transfer in organic media is interesting but is not elucidated enough.

10. The authors should provide comprehensive details of the different δ (ppm) values for H and C in the NMR spectrum of all the molecules, including their splitting in the supporting information. This information would significantly enhance understanding for the readers.

REVIEWER COMMENTS

Reviewer #1 (Remarks to the Author):

The present study addresses the synthesis of graphitic carbon nitride photoanodes for the oxidation of water and C-H functionalization. The authors have developed a new method to deposit the active semiconductor material, improving the performance with respect to the reference material. The study is interesting, but needs further work and there are many issues, which need to be clarified before the manuscript can be further considered for publication in Nature Communications. More detailed comments are provided below:

1) In the abstract, the expression "...suffers from film generation techniques." is ambiguous and does not clearly describes the motivation of the present study.

R: We thank the reviewer for this comment and agree. The related sentence has been rewritten accordingly: "..., but their photoelectrochemical (PEC) performance suffers from low charge transport capability, charge carrier recombination, and self-oxidation. High film-substrate affinity and well-designed heterojunction structures may address these issues, achieved through advanced film generation techniques."

2) The authors claim that: "All CN-based electrodes are currently limited to straightforward PEC alcohol oxidations in aqueous electrolytes.", but this is not true, and they have even cited different studies focusing on PEC water splitting (e.g.: Nat Commun 11, 4701)

R: Thank you for pointing this out. Indeed, purely inorganic reactions, such as water splitting and carbon dioxide reduction have been extensively studied, but they are not the topic of our manuscript. Instead, we focus on the potential of CN for PEC organic reactions, which are currently limited to straightforward alcohol oxidations. We now specified this, to be more precise.

3) The authors also claim that: "Here, we report a dual-layer carbon nitride (DCN) electrode with unprecedentedly high photocurrents of 910 $\mu\text{A cm}^{-2}$ at 1.23 V vs reversible hydrogen electrode (RHE)." This is not correct, the photocurrent is the current under illumination minus the dark current, which is around 500 $\mu\text{A cm}^{-2}$, according to figure 3b.

R: The reported current density values already take the "dark current" into account, which is now specified in the caption of Figure 3. The photoelectrodes have been reproduced many times, showing some batch-to-batch variations in the photocurrent. We now updated Figure 3b with a more representative measurement.

4) The spin-coating method developed is not adequate for large scale photoelectrodes. The authors should comment on the limitations of this method and how to overcome them considering the scalability of the proposed approach.

R: We agree that the scale-up of the spin coating process is a long-standing issue. but it has been extensively investigated with several different approaches. Among them, a spin-coating recipe can be well-adjusted and translated into a blade coating process for large-scale production. In fact, i a combination of blade and spin-coating has been introduced for the scalable synthesis of perovskite solar cells (Bu et al., Science 2021, 372, 1327;

Zendehdel et al., *Sol. RRL*, 2022, 6, 2100637). There, material was blade/slot-die coated and afterwards spin-coated with an antisolvent for quenching. Such a step will still require thorough optimization, but it makes a scalable spin-coating process possible.

To scale-up and translate our method to similar processes, a combination of blade and spin coating steps may be devised, where the polymer is spin coated in a poor solvent, after the deposition of the supramolecular precursor. We now discuss this in the conclusion of the manuscript.

5) The description of the synthesis provided in Figure S1 are not very clear. The steps are not clearly indicated, which leads to confusion about how the dual layer is formed.

R: Thank you for the suggestion. A much more detailed synthetic information was added to the description of Figure S1.

6) The authors claim that: “The total thickness of the DCN film is around 1.8 μm , as shown by cross-section SEM (Fig. 1d).” But from this figure, this thickness is not clearly visible.

Fig. R1-1 Topography and line profile of a photoelectrode prepared with 6 mmol/L concentration of polymer, measured by white light interferometry.

R: The topography of the DCN films was also measured by white light interferometry, resulting in 1.8 μm thickness. This result (Fig. R1-1) is now added to the supporting information.

7) The two distinct layers (insert layer and tubular layer) claimed by the authors are not visible in Figure 1.

R: The traditional CN films prepared by CVD methods are generally in nanoscale (Fig. S5). This is much thinner than the top layer of the DCN films (Fig. 1d,e & Fig. R1-2a, b). Only in the cross-section SEM images, the two layers are not clearly visible due to the significant difference of their thicknesses. We confirmed the dual-layer structure by the information from both Fig. 1 and S5. They were re-structured in Fig. R1-2. When we tried to remove the material films from the DCN electrodes, there are clearly two layers on the

substrate. The top layer is dark brown in color and can be removed by scratching, while the bottom layer is yellowish and has a stronger affinity with the substrate (Fig. R1-2d). Additionally, the morphologies of this bottom layer (Fig. R1-2f) are consistent with the CN films obtained by a typical CVD process (Fig. R1-2e, as shown by us in *Nature Communications* 2021, 12, 3224). The thickness of the DCN films is highly related to and slightly higher than the thickness of the precursor film obtained by the pre-treatment step. It further confirmed the structure of the porous top layer and thin bottom layer.

Fig. R1-2 Cross-section SEM images of a) DCN and b) CN electrodes, which were prepared by the typical CVD process (increased to 550 °C by by 1.5 °C min⁻¹, and kept at 550 °C for 3 h under nitrogen.) c) Relationship between film thickness and concentration of polymer matrix during pretreatment. d) CN photoelectrode, and DCN photoelectrodes before and after scratching the top layer. Surface view SEM image of e) CN and f) DCN electrode.

8) In Figure S4, one of the scale bars is not labelled.

R: Thank you for spotting this mistake. A scale bar has now been added.

9) The authors claim that: "...enabling diffusion similar to a solvent layer, followed by its decomposition at around 400 °C, and the formation of a carbon skeleton (Fig. S5)." But again, this is not clear from Figure S5.

R: Generally, many polymers change their phase from harder (glassy) states to soft (rubbery) states when they are heated above their glass transition temperature (T_g). For these polymers, the diffusion coefficient shows a strong change of ~2 – 5 orders of magnitude near the T_g (Tim Naylor, *in* Comprehensive Polymer Science and Supplements, 1989). Our multidisciplinary group has exploited the phase transition and decomposition

of polymer thin films for many years: We deposit thin solid polymer films, embedding chemical building blocks, such as activated amino acids. When the “solid” polymers are heated above their T_g , they soften, enabling diffusion, and allow for chemical reactions similar to a solvent (*Advanced Materials* 2022, 34, 2200359; *Advanced Materials* 2022, 34, 2108493; *Nature Communications* 2016, 7, 11844). The properties of the herein used polymers have been intensively studied in our previous publications (e.g., *Langmuir* 2022, 38, 7, 2220–2226), showing T_g s of PS 67.2 °C, S-LEC 63.5 °C, PEG ~41 °C, and PVP ~121 °C. For polystyrene, the standard polymer in our approach, the decrease in diffusion activation energies and the increase in the diffusion coefficient near T_g are more significant for larger molecules compared to smaller molecules (*Polymers* 2021, 13, 1317). The diffusion coefficient of organic molecules in PS above its T_g is $\sim 10^{-12}$ – 10^{-14} m² s⁻¹ (*Polymers* 2021, 13, 1317), which is comparable to that of DCM (10^{-10} – 10^{-11} m² s⁻¹, *Photochem Photobiol Sci* 18, 2019, 1359–1372), the solvent we used for spin coating. Thermogravimetric analysis (TGA) in Fig. S5 shows the decomposition temperature of polystyrene is around 400 °C. Next, we analyzed the C/N ratio of photoelectrodes at different temperatures (Fig. 2d). While in thermogravimetric analysis, the pure polymer begins to decompose already at 380 °C, an interaction between the precursor and the polymer could explain the upshift of polymer decomposition and observed C-C bonds in the DCN electrodes at elevated temperatures. When the temperature surpasses 500 °C, the C/N ratio of DCN decreases dramatically, suggesting that the polymer starts to decompose.

In conclusion, polystyrene softens above 67.2 °C, enabling improved diffusion of the supramolecular precursors, and then decomposes above 400 °C during the CVD process. This discussion has been carefully updated in the manuscript.

10) For the photoelectrochemical characterization, the authors should clearly indicate the geometrical area of the tested films, information about performance statistics and why a Ag/AgNO₃ reference electrode was employed for aqueous solution, since this is not the most adequate reference electrode in these conditions (Figure 3).

R: The tested film is 1 x 1 cm² on an FTO substrate with a size of 1 x 1.5 cm². Photocurrents were measured after three minutes of equilibration time. Ag/AgNO₃ and Ag/AgCl reference electrodes are used for organic and aqueous solutions, respectively. These details have now been updated in the methods part.

11) The authors claim that: “A redshift up to 800 (2)–195-nm is observed in the UV-vis spectrum (Fig. 3c), indicating that dopant and defect-related midgap states (MS) are generated in the forbidden gap of the normal CN.” But this can also arise from diffuse scattering, there is not information at all on how the optical measurements were carried out.

R: We agree that diffuse scattering could be a possible reason for the redshifted absorption. Quantifying diffuse scattering is highly difficult because many sources can contribute to it. Since absorption of longer wavelengths is only observed with the DCN and not the standard (CVD prepared) CN film samples, we can assume that the surface roughness is one of the main differences between CN and DCN samples. Yet, other factors can also cause diffuse scattering, such as thermal motion and static disorder. Therefore,

we prepared a blade-coated reference sample, resulting in a rough CN film. After the preparation of our DCN films, a CN powder remains in the crucible, which we mixed with ethylene glycol (10 mg/ml) and ground into a smooth suspension. We deposited the suspension onto an FTO substrate by doctor blading. Then, the film was dried on a hot plate at 70 °C (labeled as ‘bladed film’). This sample should have a similar chemical composition but a larger surface roughness than the standard CN film. However, no redshifted absorption was observed (Fig. R1-3), indicating diffuse scattering may not be the main reason for the redshift in the UV-vis spectrum. Meanwhile, given a strong dependence of absorption due to scattering on the wavelength ($\sim \lambda^{-4}$), absorption is stronger at shorter wavelengths. Thus, a redshift at 800 nm may not exclusively be assigned to scattering. UV-vis absorption measurements were performed in transmittance mode using a PG Instruments TG70+ UV/vis spectrometer. The measurement information and discussion on diffuse scattering were now added to the methods part.

Fig. R1-3 UV-visible absorption spectra of DCN and blade-coated CN film (inset). After preparation of the DCN electrode by CVD, the remaining CN powder in the crucible was mixed with ethylene glycol (10 mg/ml). The mixture was ground and the suspension was deposited onto an FTO substrate by doctor blading. The film was dried on a hot plate at 70 °C, yielding a strongly scattering film. UV-vis absorption measurements were performed in transmittance mode using a PG Instruments TG70+ UV/vis spectrometer.

13) The IPCE measurements should also be integrated with the solar spectrum to validate the photocurrent measurements reported in Figure 3a and b. There is no information on the specific potential at which IPCE measurements were carried out.

R: IPCE measurements were carried out at +1.23 V vs. RHE and performed with five dedicated LEDs, each with a specified wavelength (365 nm, 415 nm, 455 nm, 530 nm, 625 nm). The LEDs were connected to the LED Driver (DC2200, ThorLabs), which allows the adjustment of incident light intensity. Five data plots with different light intensities are measured for each wavelength. This information was added to the methods part. To validate the IPCE measurement, we plotted the solar spectrum and the white LED spectrum against the IPCE measurements (Fig. R1-4), now updated in Fig. 3d.

Fig. R1-4 (left) Solar spectrum plotted against the IPCE measurements of DCN and CN. (right) White light LED spectrum plotted against the IPCE measurements of DCN and CN.

14) The authors discuss about Mott-Schottky plots, but this information is not in the manuscript or SI.

R: The following Mott-Schottky plots and proposed Energy diagram (Fig. R1-5) were unfortunately missing and have now been added to the SI.

Fig. R1-5 (a) Tauc plots of DCN and CN. Mott-Schottky plots of (b) DCN and (c) CN. (d) Energy diagram of DCN.

15) Additionally, these results from Figure 3 should be validated with H₂ and O₂ gas chromatography measurements to validate that the photoelectrodes are properly working.

R: The objective of this work was to fabricate photoelectrodes for their application in non-aqueous electrolyte to enable organic transformations – herein represented by hydroxylation of benzylic position in N-aryl-tetrahydroisoquinolines (Table 1). As such, water splitting is not the main objective of our work. For stability measurements, we refer to comment 18 with Fig. R1-5.

16) The tests for organic transformation were carried out in methanol. Methanol is well-known as a hole scavenger in photoelectrochemical measurements, the authors should validate that this substrate was inert during the measurement.

R: We had similar concerns at the beginning of the project that methanol could act as a hole scavenger. To validate this, we performed the PEC reactions in different solvents, including acetone, acetonitrile, and dimethyl sulfoxide (see Table 1 in the manuscript). All these solvents give high yields. Therefore, the high performance of PEC C-H activation does not dependent on methanol. Although the thermodynamic equilibrium potential for methanol oxidation reaction (MOR) is 0.04 V vs RHE, the rate of the MOR is severely hampered by its multi-step reaction pathway. In practice, onset potentials of around +1.2 V vs RHE are required to trigger the MOR with non-noble metal-based active electrocatalysts (Adv. Mater., 2023, 35, 2211099). Given the facts that (1) no catalysts were present on the anode side during our PEC reaction and (2) the lower oxidation potential of N-aryl-tetrahydroisoquinoline of +0.22 V vs. Fc⁺/Fc (~ +0.8 vs RHE), this reaction is thermodynamically more favored compared to the oxidation of methanol. Therefore, quenching of the photogenerated holes by the amine is the more likely process compared to the oxidation of methanol.

17) Furthermore, the cyclic voltammetry curve showed in Figure 4a does not seem to be related to photocurrent and it is difficult to find the correlation with Figure 4b.

R: We thank the referee for pointing out this inconsistency between Figure 4a and 4b. There was a typo in the y-axis label, which in fact is current density and not current. The typo was corrected and the figure was updated. The cyclic voltammetry curve shown in Figure 4a is used to determine the practical oxidation potential of the THIQ substrate. Based on that, all photocurrents in Figure 4b were measured under that potential.

18) In Figure 4b, the measurement should be longer (not only 1 min) to validate that the photoelectrode is relatively stable under operation conditions.

R: To confirm the electrode stability under operation conditions, we measured the photocurrent under the same conditions as the PEC reaction for 6 h without adding THIQ substrates. Specifically, we used 0.1 M LiClO₄ in methanol as the electrolyte at +0.22 V vs. Fc⁺/Fc. The photocurrent dropped by around 40% to the end of the experiment (Fig. R1-6). Nevertheless, the stability of the photoelectrodes was sufficient for the functionalization of THIQ substrates, since one reaction only requires two hours in methanol. The surface morphologies of the photoelectrodes were characterized before and after the stability measurement. Most of the rod-like structure in the top layer could not be observed anymore, which might be caused by self-oxidation of the DCN under the photoelectrochemical conditions. This can be difficult to avoid for pure carbon nitride electrodes since it is an inherent problem of the material itself. The data about stability measurement was now added to the SI (Fig. S11).

Fig. R1-6 (a) The stability of DCN electrodes was measured under the conditions for PEC reactions (+0.22 V vs. Fc⁺/Fc, 0.1 M LiClO₄ in methanol). SEM images of the DCN electrodes (b) before and (c) after the stability test.

19) The authors carried out a comparison of the organic transformation, using Pt as a reference. But, how good is Pt to carry out this organic transformation? The authors should provide some benchmark to clearly understand the improvement with respect to the state of the art.

R: The purpose of using Pt as the anode in the dark was to compare the efficiency of the electrochemical versus the photoelectrochemical method for the oxygenation of N-aryltetrahydroisoquinoline. The reaction yield on the Pt anode is only 19%, even though the applied potential on the Pt electrode was twice higher compared to the potential applied to the DCN electrodes. These results successfully proved the concept that the PEC method offers a highly efficient pathway for organic transformation. A literature example of the electrochemical (i.e., in the dark) C-H functionalization of N-aryltetrahydroisoquinolines at the benzylic position suggests that the performance of a Pt anode is somewhat lower (30%, relative) than a carbon rod anode (Org. Biomol. Chem. 2021, 19, 8254-8258), but with *n*-Bu₄NBr-mediated oxygenation using constant current. Apart from this report, there are no examples of photoelectrocatalytic oxygenation of N-aryltetrahydroisoquinolines.

20) There are some typos to be corrected, for example there is no figure 5 in the manuscript, and References 26 and 27 are duplicated.

R: We thank the referee for pointing this out. The corresponding figure link was corrected and reference 27 was deleted.

Reviewer #2 (Remarks to the Author):

Photoelectrochemistry (PEC) is gaining momentum in recent years by providing an environmentally benign and sustainable pathway to access (bio)molecules under mild reaction conditions. The work “Metal-free photoanodes for C-H functionalization” by Savateev, Loeffler, and co-workers is a comprehensive study for the synthesis of dual-layer CN photoelectrodes with improved photoelectrochemical performance. This is true that the C-H functionalization or any other C-C OR C-N bond-forming reaction employing such photoelectrodes (PEC in general) is highly desirable from the viewpoint of sustainable chemistry and would provide an initial benchmark for future development in this field.

The CN photoelectrodes described in this work exhibit improved PEC performance with quantitative yields for most of the substrates for the said C-H functionalization with just an acceptable substrate scope. As I would have preferred an extended scope for this reaction (e.g. Dimethylanilines) to see the limitation of such a strategy. If authors have already faced such issues it should be a valuable addition to the SI of this MS.

R: We have tried another type of cross-coupling reactions using our DCN photoelectrodes, specifically, the synthesis of 4-(4-methylphenyl)-morpholine (Fig. R2-1). These results have now also been added to the SI. The DCN photoelectrode, Pt mesh, and Ag/Ag⁺ electrodes were used as working, counter, and reference electrodes, respectively. The electrolyte was 0.1 M LiClO₄ solution in methanol. Different conditions have been investigated as shown in the table. After the reaction, the solution was diluted with 5 ml double-distilled H₂O and 5 ml ethyl acetate (EtOAc). After adding 1,3-dinitrobenzene (84 mg), separated layers could be observed. The aqueous layer was extracted twice with EtOAc (5 ml). The combined organic layers were washed with brine, dried with MgSO₄, filtered, and evaporated. Then CDCl₃ (0.5 mL) was added for ¹H NMR analysis. New peaks around 7.5 ppm were observed, indicating the formation of aromatic rings. Even though the yield was low, these preliminary results show the potential of the DCN photoelectrodes for more complex C-C couplings after comprehensive optimization.

Entry	M ₁ /M ₂	AcOH	DABCO	Potential	Time	Yield
1	1:1.2	50 μ L	8.4 mg	0.58	12h	Trace
2	1:1.2	2 μ L	8.4 mg	0.58	6h	Trace
3	1:1.2	2 μ L	8.4 mg	0.22	6h	Trace
4	1:1.2	-	-	0.22	6h	-
5	1.2:1	100 μ L	-	0.22	6h	-
6	1.2:1	1.5 μ L	8.4 mg	0.22	4h	-

Fig. R2-1 Optimization of reaction conditions for the photoelectrocatalytic synthesis of 4-(4-methylphenyl)-morpholine.

On the other hand, I am a bit skeptical about the role of trace impurities (0.02% of Cu, Zn, K, Ca) in such dual-layer CN electrodes as Cu is known to cause similar oxidations. Overall, this method has the potential to provide a general platform for metal-free C-H oxidations employing photo-electro-catalysis. The method will make a nice addition to the ever-growing field of synthetic photoelectrochemistry.

R: Given the relatively low mass of the DCN film (presumably few hundreds of micrograms), we roughly estimated the Cu content in respect to the substrate in the reaction mixture to be ~ 0.005 mol. %, by taking into account (1) the volume of a carbon nitride layer (L x W x H) of 20 mm x 10 mm x 1.8 μ m with a carbon nitride gravimetric density of ~2 g cm⁻³, (2) the metal content determined by ICP in the reaction mixture and (3) the loading of N-aryltetrahydroisoquinoline (50 μ mol). This amount of metal is 200 times lower compared to, for example, the amount of homogeneous copper dinuclear complex used in a similar reaction (Chem. Eur. J. 2017, 23, 3062). If we assume that the hydroxylation of N-aryltetrahydroisoquinoline exclusively occurs due to such a low catalytic quantity of metal, then such results would be very intriguing. However, at this point we can neither exclude nor support the role of metal trace impurities as catalysts in the studied reaction.

Specific comments:

a) The scale of the optimization reaction should be mentioned in the footnote of Table 1. Did authors screen other electrodes (cathode) than Pt as Pt is an expensive metal for such oxidations?

R: We added 50 μ mol substrates in 5 ml solvent with 0.1 M LiClO₄ as electrolyte. Glassy carbon electrodes have also been used as counter electrodes during the optimization and no significant difference was observed for reaction yields. This information has been updated in the revised manuscript as suggested.

b) Also the details of experimental conditions whether in MS or SI seems to be missing. I would also prefer to see details of the experimental setup for the benefit of readers.

R: N-aryl-tetrahydroisoquinoline derivatives (THIQs) were synthesized as reported (*J. Am. Chem. Soc.* 2011, 133, 21, 8106–8109). Briefly, for N-aryl-tetrahydroisoquinoline, copper(I)diodide (190.5 mg, 1.0 mmol) and potassium phosphate (4245.4 mg, 20.0 mmol) were placed into a flask. They were evacuated and flushed with argon. 2-Propanol (20.0 mL), ethylene glycol (1241.4 mL, 20.0 mmol), 1,2,3,4-tetrahydroisoquinoline (1331.9 mg, 10.0 mmol), and iodobenzene (2.1 mL, 15.0 mmol) were added via syringes at room temperature. The mixture was heated to 85 °C for 24 h and then cooled to room temperature. Then, ethyl acetate (20 mL) and water (40 mL) were added. The aqueous layer was extracted by ethyl acetate (2×50 mL). The combined organic phases were washed with brine and dried over magnesium sulfate. After removing the solvent, the crude product was purified by column chromatography.

For PEC reactions, 106.4 mg LiClO₄ was dissolved in 5 ml of solvent (typically methanol). Then, 50 μ mol of the THIQ derivative was added to the electrolyte solution and flushed with nitrogen for 10 min. Reactions were performed with a three-electrode potentiostat (BioLogic MPG2) with a DCN photoanode as the working electrode, Ag/AgNO₃ as the reference electrode (RE), and Pt mesh as the counter electrode (CE). The setup (Fig. R1-5) was under white LED irradiation with a density of 100 mW/cm² for 2 h. To avoid high temperatures caused by light irradiation, a water-cooling jacket around the reaction vial was needed. After the reaction, the solvent was removed via rotary evaporation at room temperature. The crude product was dissolved in 500 μ l Methanol-D₄ for NMR spectroscopy with CH₂Br₂ as the internal standard.

These details and the following figure were now added to the methods part and the SI (Fig. S14) of the revised manuscript.

Fig. R2-1 The experimental setup for the photoelectrocatalytic C-H functionalization.

c) The SI should report complete synthetic procedures and characterization of the final products according to the journal guidelines. To me it is not clear whether these are crude NMRs or purified ones so maybe a detail in the SI would be helpful. Also, the NMR spectra for most of the compounds are not pure which eventually also affects the isolated yield of the reaction. So please comment. If all are crude at least one example should be confirmed via isolated yield.

R: In line with our response to comment b), details for the synthetic procedures and characterization of the final products have now been added to the methods part. Crude products were analyzed by NMR spectroscopy without further purification.

We took the functionalization of THIQ as an example to confirm the yield of the reaction. Methanol-D4 was used as the solvent to allow in-situ characterization. After 2 h, the solution was concentrated and analyzed via NMR. No substrate was observed, indicating the reaction was finished (Fig. R2-2). After a flash Al₂O₃ column chromatography, a yield of 95% was obtained for the isolated product, now shown as Fig. S15.

Fig. R2-2 ¹H NMR of the substrate and product before and after isolation.

Reviewer #3 (Remarks to the Author):

In this paper, the authors report the fabrication of CN film by using a new deposition method. The authors use the as-prepared CN electrode for C-H functionalization. The work is interesting as they introduce a new approach for fabricating new CN electrodes and the photoanode measurements in an organic medium. I believe this work can be published in Nature Communications, but the authors should carefully revise it.

1. I would like to kindly recommend the authors avoid writing extraordinary PEC performance, unprecedentedly high photocurrents, etc. It doesn't contribute to the discussion, and in some cases, it is misleading.

R: We thank the reviewer for pointing this out. We deleted 'extraordinary', 'unprecedentedly' and similar descriptions from the manuscript.

2. The LSV data (Fig 3b) should be obtained using an electrolyte solution purged with N₂/Ar.

R: O₂ reduction products, such as superoxide radicals or H₂O₂, might be observed at negative bias voltage and as such interfere with measurements. Given that the LSV measurements were conducted at a positive bias, their formation can be excluded.

3. the authors incorrectly mentioned a photocurrent density of 910 A/cm² in their study. The authors should subtract the dark current value to obtain the accurate photocurrent density.

R: The reported current density values already take the "dark current" into account, which is now specified in the caption of Figure 3. The photoelectrodes have been reproduced many times, showing some batch-to-batch variations in the photocurrent. We now updated Figure 3b with a more representative measurement.

We thank the referee for pointing out this inconsistency. The legend of Figure 3 was amended with: "These values were obtained by subtracting the "dark current density" from that measured under light irradiation."

I recommend conducting chronoamperometry measurements (light on/off) of the electrodes and providing the corresponding data.

R: Figure 4c shows the chronoamperometry experiment conducted by alternating cycles of the photoelectrode irradiation with light and in dark (0.1 M LiClO₄ in methanol), as suggested by the referee.

In addition, statements such as: "unprecedentedly high photocurrents of 910 μ A cm⁻² at 1.23 V vs. reversible hydrogen electrode" are slightly too misleading as there are reports with similar current densities.

R: As in response to comment #1, we have removed the expression "unprecedentedly high" and "extraordinary".

In addition, the photocurrent has no meaning. The Faradic efficiency and stability are more important.

R: The photocurrent may be understood as the response of the material to light excitation.

A material that demonstrates higher photocurrent under otherwise identical conditions, is able to convert photons more efficiently to generated hole-electron pairs. This is reflected in the IPCE measurements (Figure 3d). Thus, at 410 nm the DCN photoelectrode converts light 9 times more efficiently.

On the other hand, the amount of the product (n , mol) produced by electrolysis is defined by the Faraday law:

$$n = \frac{j \times S \times t}{F \times \nu}$$

where j – current density, $A\ cm^{-2}$; S – area of the electrode, cm^2 ; t – time of electrolysis, s ; F – Faraday constant, $96485\ s\ A\ mol^{-1}$; ν – number of electrons involved in the reaction.

In two experiments, using two electrodes of the same area, a larger amount of the product per time unit is obtained using a more conductive photoelectrode, i.e. the one having a higher photocurrent.

Given that no other products were formed, the oxygenation of N-aryl-tetrahydroisoquinolines is a selective process.

Oxygenation of N-aryl-tetrahydroisoquinolines was performed by running the electrolysis for 2 – 5 h. Therefore, our electrodes can survive in 0.1 M $LiClO_4$ solution for at least this period.

Does the current due to water oxidation? Or is it due to self-oxidation?

The LSV curve in Figure 3b does not show an onset of anodic current at $\sim +1.23\ V$, which could otherwise be attributed to water oxidation (assuming zero overpotential). Therefore, water oxidation at the photoelectrode may not be the main reason for current enhancement. On the other hand, decrease of the photoelectrode material resistance due to generation of charge carriers triggered by light is a more plausible explanation.

4. It is essential for the authors to conduct a stability test on the electrodes, as the stability directly impacts the efficiency of the electrodes for practical applications. Furthermore, post-characterization should be performed after the stability test.

R: The stability of the photoelectrodes was confirmed by performing an experiment for up to 6 h at $+0.22\ V$ vs. Fc^+/Fc with 0.1 M $LiClO_4$ in methanol.

To confirm the electrode stability under operation conditions, we performed chronoamperometry measurements under the same conditions as the PEC reaction for 6 h without adding THIQ substrates. Specifically, we used 0.1 M $LiClO_4$ in methanol as the electrolyte at $+0.22\ V$ vs. Fc^+/Fc . The photocurrent dropped by around 40% to the end of the experiment. Nevertheless, the stability of the photoelectrodes was sufficient for the functionalization of THIQ substrates, since one reaction only requires two hours in methanol. The surface morphologies of the photoelectrodes were characterized before and after the stability measurement. Most of the rod-like structure in the top layer could not be observed anymore, which might be caused by self-oxidation of the DCN under the photoelectrochemical conditions. This is difficult to avoid since it is an inherent problem of carbon nitride-based materials. The data about stability measurement was now added to the SI (Fig. S11).

Fig. R3-1 (a) The stability of DCN electrodes was measured under the conditions for PEC reactions (+0.22 V vs Fc^+/Fc , 0.1 M LiClO_4 in methanol). SEM images of the DCN electrodes (b) before and (c) after the stability test.

5. In Figure 3f, the authors measured the lifetime at 530 nm, while the emission maximum is between 450-460 nm (Fig. 3e). This discrepancy should be explained.

R: The spectra presented in Figure 3e were acquired using a JASCO FP-8300 fluorescence spectrometer (excitation at 365 nm) that only operates in a steady-state mode. In this case, the excitation beam was obtained by passing light from a Xe lamp through a monochromator. Time-resolved decay of fluorescence intensity shown in Figure 3f was acquired using a picosecond laser diode (375 nm, spectrometer PicoQuant TCSPC FluoTime 250). Using TCSPC-TRES in steady-state mode, the fluorescence spectrum of a DCN electrode was acquired upon excitation at 375 nm and is shown in Fig. R3-2. There, a fluorescence maximum at ~ 530 nm was observed. The difference in the fluorescence maxima in Fig. 3e and Fig. R3-2 might be due to the variation in the excitation wavelengths and light intensities. This was now included in the SI.

Fig. R3-2 Steady-state fluorescence spectrum of a DCN electrode acquired using a picosecond laser diode (excitation at 375 nm) and spectrometer PicoQuant TCSPC FluoTime 250 operating in steady-state mode.

6. The authors claimed that the activity of the DCN sample is due to the formation of C-C bonds. Consequently, these bonds should be visible in the FTIR spectrum of DCN (Fig. S6). Additionally, they should be observed in the ^{13}C solid-state NMR of DCN (Fig. 2e, Fig. S7).

R: We have not observed any significant changes, since we expect only very little remaining carbon from the synthesis process (see Fig. S5, TGA of polymer). FTIR measurements for C-C bonds are typically subtle (more qualitative in the range of $\sim 600 - 1400\text{ cm}^{-1}$). Thus, FTIR is typically suited for more prominent bonds and/or material changes. The lower sensitivity limit of ^{13}C NMR, especially in solid state, is about 10 %. Since the C-C bond content is expected below this value, the corresponding signal will merge with the baseline. However, by XPS analysis, even after 5 h of argon etching, the C-C peak was still present (Fig. 2c), indicating that C-C bonds are actually part of the DCN film structure, introduced during the process.

7. The authors presented three methods for constructing a CN film: doctor-blading, spin coating, and dip-coating, using DCM as the solvent. However, while DCM might be a suitable solvent for spin coating, it may not be suitable for other methods, such as doctor blading. Therefore, I recommend the authors explore alternative solvents such as ethylene glycol, and for fair comparison (which I don't think is needed), they should adopt the best method from the literature.

R: Ethylene glycol was used in the beginning for doctor blading. No significant differences in the PEC performance were observed when we switched the solvent from ethylene glycol to DCM. Therefore, we showed the results with DCM to make the conditions more comparable with those for spin coating.

8. The authors should also consider moderating their statement in the abstract: "...but their photoelectrochemical (PEC) performance suffers from film generation techniques." in the last years, many methods have been introduced to develop excellent CN films. The PEC performance may suffer from self-oxidation of CN, inefficient charge separation, etc.

R: We thank the reviewer for this comment and agree. The related sentence has been rewritten accordingly: "..., but their photoelectrochemical (PEC) performance suffers from low charge transport capability, charge carrier recombination, and self-oxidation. High film-substrate affinity and well-designed heterojunction structures may address these issues, achieved through advanced film generation techniques."

9. Mechanistic investigation for the C-H functionalization using DCN film is missing. The charge transfer in organic media is interesting but is not elucidated enough.

R: Considering published data (Synlett 2019, 30, 2077-2080; Org. Biomol. Chem., 2021,19, 8254-8258) the photoelectrochemical hydroxylation mechanism of N-aryl-tetrahydroisoquinolines is outlined in Fig. R3-3 and now summarized in the Supplementary Discussion 1.

Excitation of the DCN photoanode by light converts carbon nitride into an excited state. The bias voltage of +0.22 V facilitates the separation of charges by extracting electrons. Photogenerated holes oxidize N-aryl-tetrahydroisoquinoline to the corresponding

iminium cation ($2e^-/H^+$ process), which upon nucleophilic attack of HO^- is converted into the product. Iminium cations are ubiquitous intermediates in photoredox catalysis and were also postulated in photocatalytic functionalization of N-aryl-tetrahydroisoquinolines by carbon nitrides (J. Am. Chem. Soc. 2010, 132, 5, 1464–1465; Adv. Synth. Catal., 354, 1909-1913). Due to the constant bias potential of only +0.22 V vs. Fc⁺/Fc, compared to a constant current of 5 mA (Synlett 2019, 30, 2077-2080), the aminoalcohol 2 is selectively obtained instead of the amides. HO^- species are replenished upon water reduction at the Pt cathode. Although the scope of the reaction was investigated in methanol, screening of reaction conditions revealed that the yield of 2a was high (86%), when the reaction was performed in wet acetonitrile and acetone (Table 1, entries 1,2).

Fig. R3-3 Proposed mechanism of photoelectrochemical oxygenation of N-aryl-tetrahydroisoquinolines.

10. The authors should provide comprehensive details of the different δ (ppm) values for H and C in the NMR spectrum of all the molecules, including their splitting in the supporting information. This information would significantly enhance understanding for the readers.

R: The details for the NMR spectra were now given in the SI. The yield for compound 2f was corrected to 49%.

REVIEWER COMMENTS

Reviewer #1 (Remarks to the Author):

The authors have convincingly addressed all raised issues, now the manuscript can be accepted for publication.

Reviewer #2 (Remarks to the Author):

After going through the amended version, I can accept this MS after some further revision. I would have preferred to see the limitation and generality of this method (already mentioned) in terms of scope as only a few activated amines have been used. Spectroscopic data should be presented in a more uniform manner (journal-specific style) and besides NMR data mass of the final product should also be recorded (even GCMS up to one digit would suffice) but more according to journal guidelines even if these known compounds. Overall, most of the referee's criticisms have been satisfactorily addressed, including specific corrections/additions in the MS and SI that I believe were required.

Reviewer #3 (Remarks to the Author):

The authors addressed most of the reviewers' remarks and the paper is significantly improved. I have only one remark:

1. The Faradic efficiency should be provided. The photocurrent can be due to the self-oxidation of the electrode. They need to calculate the charge they passed with the product formation. This is the true value to evaluate the performance.

Reviewer #1 (Remarks to the Author):

The authors have convincingly addressed all raised issues, now the manuscript can be accepted for publication.

R: We appreciate referee's positive feedback and recommendation to publish our work in *Nature Communications*.

Reviewer #2 (Remarks to the Author):

After going through the amended version, I can accept this MS after some further revision. I would have preferred to see the limitation and generality of this method (already mentioned) in terms of scope as only a few activated amines have been used.

R: Although we have confirmed the possibility of using DCN electrodes to perform cross-dehydrogenative couplings, substantial optimization of the catalytic system is required to make the approach synthetically useful. This will be part of our future investigations, but is beyond the scope of this manuscript.

Spectroscopic data should be presented in a more uniform manner (journal-specific style) and besides NMR data mass of the final product should also be recorded (even GCMS up to one digit would suffice) but more according to journal guidelines even if these known compounds.

R: Thank you for this comment. We have performed all necessary NMR characterizations for compounds **2a–f** in a uniform style in the supporting information and included citations to earlier reports. Unfortunately, since all executing authors meanwhile have moved to new institutions around the world and the specific labs have been suspended, we cannot perform the MS analyses for compounds 2b-2f.

Overall, most of the referee's criticisms have been satisfactorily addressed, including specific corrections/additions in the MS and SI that I believe were required.

R: Thank you very much for this supportive comment.

Reviewer #3 (Remarks to the Author):

The authors addressed most of the reviewers' remarks and the paper is significantly improved. I have only one remark:

1. The Faradic efficiency should be provided. The photocurrent can be due to the self-oxidation of the electrode. They need to calculate the charge they passed with the product formation. This is the true value to evaluate the performance.

R: Considering a photocurrent density of 500 $\mu\text{A cm}^{-2}$ with 1 cm^2 of the photoelectrode immersed into the electrolyte, a 2 h duration of the photoelectrocatalytic experiment, a 50 μmol loading of **1a-f**, and the yields for **2a-f**, the Faradaic efficiency was determined to be ~67% (**2a-d**), ~60% (**2e**), and ~33% (**2f**).

This has now been included in the manuscript.

The Faradaic efficiency (F.E.) was calculated according to the following equation:

$$F.E. = \frac{n_p}{n_e} \times 100\% = \frac{n_r \times \eta}{\left(\frac{N_e \times j \times S \times t}{q_e \times N_A} \right)}$$

with n_p – amount of **2a-f** generated in the photoelectrocatalytic experiment, mol; n_e – amount of electrons passed through the PEC cell, mol; n_r – loading of **1a-f**, mol; η - chemical yield of **2a-f**, %; N – number of electrons involved in the chemical transformation, i.e. $N_e = 2$ according to the proposed mechanism; q_e – elementary charge, 1.6×10^{-19} C; j – current density, A cm^{-2} ; S – area of the photoelectrode, cm^2 ; t – time of the photoelectrochemical experiment, s; N_A – Avogadro constant, $6.02 \times 10^{23} \text{ mol}^{-1}$.

REVIEWERS' COMMENTS

Reviewer #2 (Remarks to the Author):

The MS can be accepted now based on the revision and replies from the authors.

Reviewer #3 (Remarks to the Author):

The authors addressed the last comment and I am happy to recommend this nice work acceptance.